# Higher Collective Responsibility, Higher COVID-19 Vaccine Uptake, and Interaction with Vaccine Attitude: Results from Propensity Score Matching

**DOI:** 10.3390/vaccines10081295

**Published:** 2022-08-10

**Authors:** Jianwei Wu, Caleb Huanyong Chen, Hui Wang, Jinghua Zhang

**Affiliations:** 1School of Business, Macau University of Science and Technology, Macao 999078, China; 2Nursing and Health Education Research Centre, Kiang Wu Nursing College of Macau, Macao 999078, China; 3Education Department, Kiang Wu Nursing College of Macau, Macao 999078, China

**Keywords:** vaccine hesitancy, collective responsibility, COVID-19 vaccine, propensity score matching, theory of planned behavior

## Abstract

Providing both personal and social benefits, vaccination may be motivated by collective responsibility (CR). Some previous studies have indicated the relationship between CR and vaccination but could not exclude confounding bias and had little knowledge about the boundary conditions. This study aimed to examine the association between CR and COVID-19 vaccine uptake and its boundary conditions in an extended version of the theory of planned behavior. A cross-sectional survey with 608 participants from six tourism satellite industries in Macao was conducted from 28 July 2021 to 20 August of 2021. Respondentss in CR-lower and CR-higher groups were 1:1 paired using propensity score matching (PSM) to control the potential confounding factors. Results showed participants in the CR-higher group reported significantly higher COVID-19 vaccine uptake than those in the CR-lower group (64.7% vs. 49.7%, *p* = 0.005). Multivariate logistic regression results indicated a positive association between CR and COVID-19 vaccine uptake (*p* = 0.012, OR = 2.070, 95% CI= 1.174 to 3.650) and its interaction effect with COVID-19 vaccine attitude (*p* = 0.019, OR = 0.922, 95% CI = 0.861 to 0.987). Spotlight analysis further illustrated that CR was more effective among individuals with a more negative COVID-19 vaccine attitude. These findings may help promote understanding of vaccine hesitancy, and hence optimize vaccination communication strategies during the COVID-19 pandemic.

## 1. Introduction

The coronavirus disease 2019 (COVID-19) pandemic has deeply impacted human lives in multiple aspects and resulted in above 6.3 million deaths across 220 countries as of 9th July 2022 [1]. Reliable evidence indicates that vaccination remains the safest and most effective strategy to prevent COVID-19 infections, hospitalizations, and death [2]. However, hesitancy and reluctance about COVID-19 vaccine injections were still observed among some populations worldwide [3,4]. Vaccine hesitancy was listed among the top ten threats to global health by the WHO [5]. Many previous studies have applied health behavior theories, theory of reasoned action (TRA), or theory of planned behavior (TPB) in particular [6,7], attempting to address this problem. In addition, since vaccination not only provides a direct personal benefit to the vaccinated individual but also indirectly provides a social benefit to the unvaccinated such as children and older adults, it can be considered a prosocial behavior and is associated with collective responsibility (defined as the willingness to protect others by one’s own vaccination by means of herd immunity) [8]. A systematic review found that 29 out of 470 studies identified social benefit as a significant influence on influenza vaccine uptake, thus social benefit (e.g., collective responsibility) was added as one of the psychological determinants for vaccine uptake in the extended version of TPB [9]. Collective responsibility, or social accountability, was also proposed by behavioral scientists as an important motivator in COVID-19 vaccination promotion [10,11].

TRA, introduced by Fishben [12], posits that if an individual evaluates a behavior as positive (attitude), and if the individual believes that he or she is expected to perform the behavior (subjective norm), the individual will have more intention to perform the behavior. Subsequently, TRA was extended to TPB by adding the “Perceived Behavioral Control” component (a person’s perception of the ease or difficulty of performing the behavior of interest) to better predict actual behavior [13]. Therefore, according to TPB, attitude, subjective norm, and perceived behavioral control are the three core components that together shape an individual’s intention to take actions, such as vaccination. A meta-analysis also demonstrated that these three components were significant predictors of vaccination intentions [14]. However, most current research on COVID-19 vaccination is overwhelmingly based on evidence of people’s vaccination intention rather than vaccine uptake. Although there is a causal relationship between intention and behavior in most health-related models [15], some studies pointed out that there is a gap between intention and behavior, vaccination intention does not always transfer to vaccine uptake [16]. 

Collective responsibility was introduced to study health-related behaviors (e.g., reducing sexual transmission of HIV) early [17,18] and was included as a psychological antecedent of vaccination in both the 5C model and the extended 7C model of vaccination intentions recently [19,20]. An increasing number of empirical studies support that people scoring higher in collective responsibility show greater vaccination intentions and report higher vaccine uptake [19,20,21,22]. However, previous studies have little knowledge about the boundary conditions (referring to the ‘who, where, when’ aspects of a theory) of collective responsibility affecting vaccination. Böhm and Betsch [23] proposed that promoting prosocial vaccination (e.g., collective responsibility) may be more effective among individuals with lower vaccination motivation but called for empirical study. This calls into question whether collective responsibility is an independent predictor of vaccine uptake or whether it interacts with other predictors in the extended version of TPB. Additionally, vaccination willingness or behaviors are influenced by many factors [3,9,24,25,26,27,28], and there are potential self-selection or confounding biases when studying vaccination intention or behaviors based on observational samples or data. For example, someone who has higher collective responsibility may also have a more positive attitude or higher subjective norm on vaccination. Studies based on the traditional regression analysis method often cannot address the concerns about self-selection bias or unobservable confounding variables as discussed above [23], whereas the propensity score matching (PSM) method is an effective alternative method to reduce the confounding bias in observational studies [29]. 

Macao Special Administration Region (SAR) of China is in the western Pearl River Delta by the South China Sea. Featured as a crowded urban setting with high buildings, Macao has a population of about 680,000 and is the most densely populated region in the world [30]. Macao has become a major international resort city and a top destination for gambling tourism, which contributes to 60% of the local GDP and 70% of the local tax revenue. The total number of workers in the six tourism satellite industries (gaming, retail trade, food and beverage, hotel, passenger transportation, and travel agency services) was around 203,000 in 2019, accounting for nearly 50% of the working population in Macao [31]. Up to 7 July 2022, 89.1% of the population of Macao has been fully vaccinated in comparison with the global rate of 61.3% [32]. Workers in tourism satellite industries may generally have a stronger willingness to end the COVID-19 pandemic due to the severe negative impacts on the tourism industry. The government of Macao (SAR) has adopted a variety of strategies to promote COVID-19 vaccination, including promoting residents’ collective responsibility [33]. Therefore, the case of tourism satellite industries in Macao has provided a good research opportunity to study whether CR can play a positive role to reduce vaccine hesitancy. Adopting a PSM technique to reduce self-selection or other potential confounding bias in observational data, this study aims to explore the role of CR in vaccination decision making based on the model of TPB, as well as its boundary conditions. 

## 2. Materials and Methods

### 2.1. Participants and Study Design

Ethical approval was obtained from the School of Business of Macau University of Science and Technology (Reference number: MUST/MSB/2021/03). A cross-sectional survey using a structured questionnaire was conducted among workers in the six tourism satellite industries (gaming, retail trade, food and beverage, hotel, passenger transportation, and travel agency services) in Macao. The inclusion criteria were full-time adult employees in the studied industries who consented to participate in this study and could answer an online e-questionnaire in the Chinese language. The participants were recruited using convenience sampling. The questionnaire with a written consent form was posted on an online survey platform (Google Forms). A poster with a QR code and a website link of the survey were created, respectively. Both emails and printed posters were used to disseminate the survey information with the assistance of the companies’ human resources departments or heads of the industry associations. 

According to the sampling formula n=Z∝/22P^(1−P^)Nδ2(N−1)+Zα/22P^(1−P^) (α = 0.05, δ = 0.05, *p* = 0.5), the minimum sample size was around 400. A quality control question (what year is this year?) was set to detect inattentive samples. Data were collected from 28th July to 20th August of 2021 after a 47-sample pilot study. The progress of the survey was monitored daily. Data were exported from the online survey platform and checked for errors: (i) removed those who did not complete the questionnaire, and (ii) excluded those who answered the quality control question incorrectly. Statistical analyses were performed using SPSS (Version 28,SBAS Ltd., Hong Kong) for Windows.

### 2.2. Measures

The self-administered questionnaire included (i) COVID-19 vaccine uptake, (ii) COVID-19 vaccine attitude, subjective norm, and perceived behavioral control, (iii) collective responsibility, and (iv) socio-demographic characteristics. In addition, (ii) was designed based on the extended theory of TPB and previous studies. Seven experts in public health were asked to evaluate the content validity, calculating the Item-level Content Validity Index (I-CVI) and Scale-level Content Validity Index (S-CVI). When the experts are no less than 6, the content validity is considered to be good if I-CVI ≥ 0.78 and S-CVI ≥ 0.90 [34]. The exploratory factor analysis (EFA) using principal components analysis (PCA) was performed to test the construct validity. The internal consistency reliability was assessed with Cronbach’s alpha, and it is good when factor analysis is applied and Cronbach’s alpha is between 0.70 and 0.95 [35].

#### 2.2.1. COVID-19 Vaccine Uptake (VU)

COVID-19 VU was reported by a single item: “Have you received COVID-19 vaccine?” The Item was rated on a 5-point scale: 1 = not willing to be vaccinated, 2 = undecided whether to vaccinate, 3 = not yet, but planning to vaccinate, 4 = one dose has been received, and 5 = two doses have been received. Participants answering 1–3 were defined as unvaccinated against COVID-19; others answering 4 or 5 were defined as vaccinated. 

#### 2.2.2. COVID-19 Vaccine Attitude (VA)

COVID-19 VA was measured using a belief-based measure [36]. Participants were presented with four paired items of behavioral beliefs and their corresponding outcome evaluation. Behavioral beliefs include “I think getting the COVID-19 vaccine (i) can protect myself from COVID-19 infection, (ii) has additional benefits, such as accessibility, special holidays, etc., (iii) has a safety hazard, (iv) has side effects. The first two beliefs were positive attitudes, and the others were negative attitudes. The corresponding outcome evaluation items are (i) “I value the protection of COVID-19 vaccine”, (ii) “I value the additional benefits, such as accessibility, special holidays, etc.” (iii) “I am scared of safety hazard from COVID-19 vaccine”, (iv) “I am scared of the side effects from COVID-19 vaccine”. Each item was scored on a 5-point unipolar scale ranging from 1 to 5. 

A belief-based measure of the COVID-19 VA was calculated as the summation of the four products of the paired behavioral beliefs and outcome evaluations. A higher VA score indicates a more positive attitude. I-CVI of this scale ranged from 0.86 to 1.00, and S-CVI was 0.93. A two-factor solution explaining 63.26% of the total variance (36.96% for positive items, 26.30% for negative items) was obtained by EFA in our sampling (KMO = 0.724, Bartlett’s test χ^2^ = 1858.616, *p* < 0.001). Cronbach’s alpha of the two factors was 0.69 (around 0.70) and 0.86. Thus, the scale has sufficient validity and reliability.

#### 2.2.3. Subjective Norm (SN)

SN was also measured using a belief-based measure [36]. Four referents were used in the measure of normative beliefs (i.e., my family, my employer, medical professionals I trust, and associations I trust). Participants were asked to indicate the degree that each referent recommend them to take the COVID-19 vaccine, and the degree that they were motivated to comply with each referent’s recommendation. Each item was scored on a 5-point unipolar scale ranging from 1 to 5. A belief-based measure of SN was calculated as the summation of the four products (normative belief × motivation to comply), with a higher score indicating higher SN. Both I-CVI and S-CVI of the scale were 1.00. One factor explained 64.74% of the total variance using EFA in our sampling (KMO = 0.828, Bartlett’s test χ^2^ = 3740.009, *p* < 0.001), and the Cronbach’s alpha was 0.92. Thus, the scale has sufficient validity and reliability.

#### 2.2.4. Perceived Behavioral Control (PBC)

PBC was measured with a single item asking participants to report their perception of the ease or difficulty in getting COVID-19 vaccination [36]. The item was rated on a 5-point scale (1 = very difficult, 5 = very easy).

#### 2.2.5. Collective Responsibility (CR)

CR was measured by the three-item subscale of the 5C scale, a valid brief measure assessing psychological antecedents of vaccination, and the subscale’s Cronbach’s alpha was 0.71 with sufficient reliability [19]. Items were rated on a 5-point scale from 1 = strongly disagree to 5 = strongly agree. One item (‘When everyone is vaccinated, I don’t have to get vaccinated too’) was reversed to be in line with this scoring. A composite variable was computed by averaging the three items, with a higher average score indicating higher CR. The score of the composite variable ranged from 1.67 to 5.00 (mean = 3.72, SD = 0.64). Participants with a score lower than the average were sorted into the CR-lower group, and others were sorted into the CR-higher group. 

#### 2.2.6. Socio-Demographic Characteristics

These consisted of gender, age, educational level, marital status, monthly income, whether living with older adults or children or not, and working industries.

### 2.3. Statistical Analysis

Propensity score matching was performed in this study. Using multivariable logistic regression, a propensity score of each participant being in the CR-higher group was predicted and a 1:1 matching of the scores was performed [37]. The match tolerance was set at 0.02. The following covariates were used for the PSM analysis: COVID-19 VA, SN, PBC, gender, age, educational level, marital status, monthly income, whether living with older adults or children or not, and working industries. Balance-check of the covariates was conducted to confirm the validity of applying this method, using both *p*-value and standardized mean difference (SMD) as criteria. With potential confounding factors controlled, the association between collective responsibility and COVID-19 vaccine uptake was estimated with multivariable logistic regression analysis. The odds ratio (OR) and 95% confidence interval (CI) were calculated. In the multivariable logistic regression analysis, we also examined the interaction effect between predictors and interpreted the interactions with a spotlight analysis [38]. Significance was accepted at the *p*-value < 0.05 for all calculations.

## 3. Results

### 3.1. Participant Characteristics

A total of 620 questionnaires were collected and 608 (98.06%) were valid for analysis, with 125 (20.56%) from travel agency industry, 167 (27.47%) from food and beverage industry, 178 (29.27%) from gaming industry, and 138 (22.70%) from the other three industries (retail trade, hotel, and passenger transport). Age ranged from 18 to 72 (mean = 38.26, SD = 10.36). As shown in Table 1, more than half of the participants (62.5%) have received COVID-19 vaccine. Figure 1 shows that 268 participants were assigned to the CR-lower group and 340 were assigned to the CR-higher group according to the criteria defined in the 2.2 Measures section, and 173 pairs were selected through PSM treatment according to the criteria defined in the 2.3 Statistical Analysis section.

### 3.2. Baseline Covariates after Propensity Score Matching

The comparisons of baseline characteristics before PSM and after PSM are presented in Table 2. Before PSM treatment, there were significant differences (*p* < 0.05; SMD > 0.10) among gender, age, working industries, COVID-19 VA, SN, and PBC between the CR-lower and the CR-higher groups. After PSM treatment, there were no significant differences regarding the above-mentioned characteristics between these two groups (*p* > 0.05; SMD ≤ 0.10).

### 3.3. The Association of Collective Responsibility and COVID-19 Vaccine Uptake

Multivariate logistic regression analysis in Table 3 shows that the significant independent predictors of COVID-19 vaccine uptake were CR (*p* = 0.012, OR = 2.070, 95% CI = 1.174 to 3.650), COVID-19 VA (*p* < 0.001, OR = 1.149, 95% CI= 1.086 to 1.216) and PBC (easy or very easy = 1) (*p* = 0.019, OR = 5.636, 95% CI = 1.330 to 23.882). The model successfully classified 78.3% of cases overall (R2N = 0.465). The results indicated that participants who had higher CR, more positive COVID-19 VA, and perceived getting COVID-19 vaccination as easier would have higher COVID-19 vaccine uptake. Table 3 also shows CR and COVID-19 VA had an interaction effect on COVID-19 vaccine uptake (*p* = 0.019, OR = 0.922, 95% CI = 0.861 to 0.987).

To better understand the interaction effect between CR and COVID-19 VA, we conducted a spotlight analysis to further interpret the effect of CR on COVID-19 vaccine uptake among participants with more positive (one standard deviation above mean) and more negative (one standard deviation below mean) COVID-19 VA. As shown in Figure 2, CR was moderated by COVID-19 VA, and it was more effective among individuals who had more negative COVID-19 VA. It further illustrates that the CR–COVID-19 vaccine uptake relationship was stronger among participants who had a more negative COVID-19 VA.

## 4. Discussion

### 4.1. Higher Collective Responsibility Predicts Higher COVID-19 Vaccine Uptake

As shown in Table 2, participants in the CR-higher group reported a significantly higher COVID-19 vaccine uptake than those in the CR-lower group (64.7% vs. 49.7%) after PSM. Logistic regression also indicated a positive association between CR and COVID-19 vaccine uptake. Previous studies have already demonstrated that collective responsibility was positively associated with vaccination intentions or vaccine uptake. For example, an online survey in China found participants’ COVID-19 vaccination intention was positively associated with CR [22]; a survey in Hong Kong found nurses scoring higher on CR showed greater influenza vaccine uptake and COVID-19 vaccination intention [21]; another survey in the UK found that a lower sense of CR independently predicted a lack of uptake of influenza, pneumococcal, and shingles vaccine among older adults [39]. Some other studies also found similar results [40,41]. However, vaccine uptake is associated with many factors [3,9,24,25,26]. The present study illustrated that people who have different CR may also have significant differences in terms of COVID-19 VA, SN, and PBC before PSM (Table 2), which may lead to confounding bias. Moreover, previous studies often did not exclude these potentially influential factors as alternative explanations. Compared with these studies, the present study reduced the confounding bias of participants’ COVID-19 VA, SN, PBC, and the socio-demographic characteristics shown in Table 2 with the method of PSM. Thus, this study added better evidence to current research. 

Additionally, Table 3 shows that COVID-19 VA and PBC are also positively associated with COVID-19 vaccine uptake, although the association of SN is not significant (*p* = 0.080). The results were similar to many previous studies [42,43,44,45]. It indicates that the theory of planned behavior can be well applied to study people’s COVID-19 vaccine hesitancy and behaviors. It also proves the necessity of controlling these covariates with PSM in this study.

### 4.2. Collective Responsibility Is More Effective among Individuals Having More Negative COVID-19 Vaccine Attitude

This study verified that CR and COVID-19 VA had an interaction effect on COVID-19 vaccine uptake, and the spotlight analysis illustrates that the CR –COVID-19 vaccine uptake relationship was stronger among participants who had a more negative COVID-19 VA. On the contrary, participants with a positive vaccine attitude can be motivated to vaccinate by the benefits of vaccines (e.g., protecting themselves), regardless of whether CR is high or low. Figure 2 illustrates that the COVID-19 vaccination rates among participants with positive COVID-19 VA were similarly high in the CR-lower group and the CR-higher group. A meta-analysis study also found that attitude towards vaccine was the strongest predictor of vaccination intention [14]. Conversely, those with negative attitudes towards COVID-19 vaccine may have lower motivation to vaccinate for direct personal benefits, thus whether they value an indirect social benefit is important. The finding supports the hypotheses proposed by Böhm and Betsch [23] that prosocial vaccination (e.g., collective responsibility) may be more effective among individuals with lower vaccination motivation. In addition, it further answered that CR is a predictor of vaccine uptake and will also interact with vaccine attitude in the extended version of TPB.

### 4.3. Implications

The findings of the present study provide theoretical and practical implications. To begin with, to our knowledge, this is the first study aimed to examine the association between CR and COVID-19 vaccine uptake based on the PSM method. The key findings that higher CR is associated with higher COVID-19 vaccine uptake and that it is more effective among individuals having more negative COVID-19 vaccine attitude emerged, uniquely contributing to our understanding of people’s COVID-19 vaccine hesitancy and behaviors.

These findings may provide policymakers, health educators, and practitioners with insights into vaccination communication. Firstly, CR is an important motivator in COVID-19 vaccination promotion and calls for strategies. People’s vaccine attitudes and behaviors are not immutable. One study found that 60% of those who initially reported some level of hesitancy eventually got vaccinated [46]. Böhm and Betsch [23] proposed that CR can be promoted through communication campaigns regarding awareness of herd immunity and community protection. For example, Macao had been doing well and maintaining several months without new COVID-19 cases before May 2021 [47] despite frequent outbreaks in neighboring cities [48]. To respond to the unprecedented threat, the government of Macao SAR took a series of measures to encourage vaccination, especially, city-wide slogans communicating knowledge about community protection like: ‘we are all guardians, get vaccinated for yourself and others’ and ‘vaccination together to build an immune barrier’ [33]. These promotions may have helped to motivate people’s CR to get COVID-19 vaccination. The vaccination rate in Macao rapidly increased from 15.22% on 27th May to 35.70% on 30th June [32]. Moreover, the government of Macao SAR is currently trying its best to motivate the public’s CR to take measures (e.g., several rounds of city-wide COVID-19 testing) to cope with the latest outbreak involving more than 1000 locally infected cases since 18 June 2022 [33].

Secondly, the present study shows that the CR–COVID-19 VA relationship was stronger among participants who had more negative COVID-19 VA. Hence, educators and practitioners perhaps should focus on CR promotions among people with less positive COVID-19 VA. For example, in Canada, parents using the internet to search for vaccination information often have a negative perception of vaccine risks and safety [49].

### 4.4. Limitations and Future Research

The present study had some limitations. Firstly, while generally consistent with randomized clinical trials when applied appropriately, the PSM method has its own limitations. Some data information may be lost during the process of propensity score matching because observations without suitable matches are excluded in the final estimation [50]. PSM method may help to alleviate the self-selection bias and the omitted variable bias, but cannot fully address these concerns.

Secondly, this study is a cross-sectional design and the effectiveness of examining causality is limited. Awareness of collective responsibility may vary during stages of the COVID-19 pandemic [51]. Further time sequence studies (e.g., prospective cohort study) are needed to verify the reliability of these results.

Thirdly, the case of Macao may not have high representativeness worldwide, since Macao is an Asian city with the highest population density in the world and its economy heavily depends on tourism. Meanwhile, a US-based study found that prosocial concerns promote vaccination against COVID-19 more in sparsely rather than densely populated areas [52]. The findings of this study should be generalized with caution.

As for study directions in the future, a further investigation of the measurement of CR may help to deepen the understanding of the boundary conditions of CR. As Böhm and Betsch [13] proposed, promoting CR may also be more effective among individuals who have great CR but lack knowledge regarding herd immunity and community protection. Exploring more boundary conditions of CR (e.g., is the role of CR the same in different cultural backgrounds?) and CR communication intervention studies may further demonstrate the application value.

## 5. Conclusions

Applying the PSM method to control potential confounding bias among observational data, this study found that higher CR predicted higher COVID-19 vaccine uptake. Participants in the CR-higher group in this study reported significantly higher COVID-19 vaccine uptake than those in the CR-lower group (64.7% vs. 49.7%) after PSM treatment. The significant interaction effect of CR and COVID-19 VA indicates that the association is especially strong among those having a negative COVID-19 vaccine attitude. The empirical evidence found in this study hence may help optimize communication strategy to enhance or maintain sustainable COVID-19 vaccine uptakes.

## Figures and Tables

**Figure 1 vaccines-10-01295-f001:**
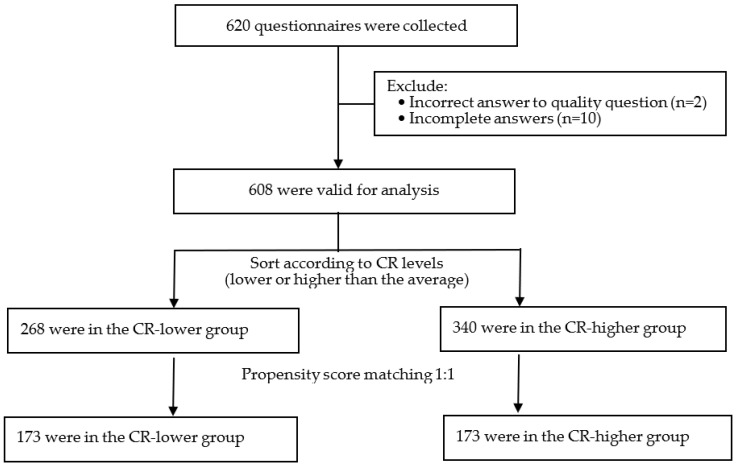
Flowchart of data processing.

**Figure 2 vaccines-10-01295-f002:**
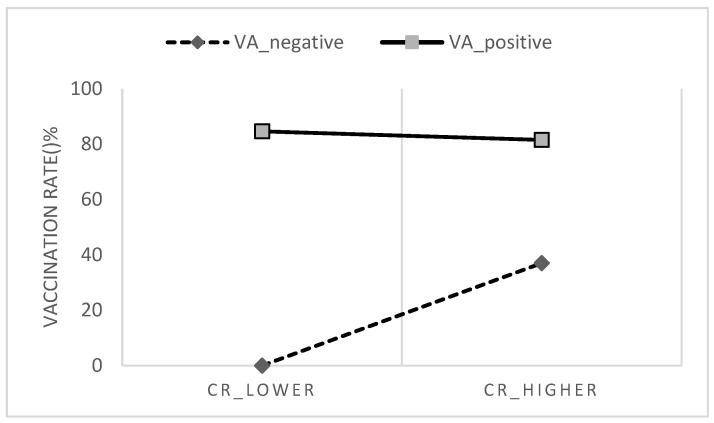
Collective responsibility moderated by vaccine attitude.

**Table 1 vaccines-10-01295-t001:** Demographic characteristics (*n* = 608).

Variables	Count	%
**Gender**		
Male	222	36.5
Female	386	63.5
**Education**		
High school or below	284	46.7
Diploma or above	324	53.3
**Marital status**		
Single	236	38.8
Married	372	61.2
**Living with older adults or children**		
No	217	35.7
Yes	391	64.3
**Monthly income (in local currency MOP)**		
**≤** **10,000**	98	16.1
**10,001** **∼** **20,000**	237	39.0
**20,001** **∼** **30,000**	198	32.6
**30,001** **∼** **30,000**	49	8.1
**≥** **40,001**	26	4.2
**Working industries**		
Travel agency	125	20.6
Gaming	167	27.5
Food and beverage	178	29.3
Others	138	22.7
**Receiving COVID-19 vaccine**		
Not willing to be vaccinated	38	6.3
Undecided whether to vaccinate	109	17.9
Not yet, but planning to vaccinate	81	13.3
1 dose has been received	86	14.1
2 doses have been received	294	48.4

**Table 2 vaccines-10-01295-t002:** PSM to balance the participants’ characteristics between the CR-lower and CR-higher groups.

Characteristics	Before PSM				After PSM			
	CR-Lower Group	CR-Higher Group	SMD	*p*-Value	CR-Lower Group	CR-Higher Group	SMD	*p*-Value
No. of participants	268	340			173	173		
Age, mean ± SD	36.73 ± 9.53	39.46 ± 10.84	0.26	0.001	39.67 ± 9.75	39.64 ± 10.90	<0.01	0.974
COVID-19 VA, mean ± SD	−2.25 ± 12.94	8.20 ± 13.22	0.74	<0.001	2.23 ± 10.93	2.82 ± 11.67	0.05	0.628
SN, mean ± SD	49.25 ± 22.72	67.19 ± 23.93	0.71	<0.001	56.45 ± 23.00	55.54 ± 21.17	0.04	0.703
PBC			0.26	<0.001			0.04	0.808
Difficult or very difficult	22 (8.2%)	9 (2.6%)			7 (4.0%)	9 (5.2%)		
Moderate	98 (36.6%)	62 (18.2%)			52 (30.1%)	48 (27.7%)		
Easy or very easy	148 (55.2%)	269 (79.1%)			114 (65.9%)	116 (67.1%)		
Gender			0.71	0.029			0.01	0.819
Male	85 (31.7%)	137 (40.3%)			58 (33.5%)	56 (32.4%)		
Female	183 (68.3%)	203 (59.7%)			115 (66.5%)	117 (67.6%)		
Education			0.03	0.532			0.04	0.452
High school or below	129 (48.1%)	155 (45.6%)			92 (53.2%)	85 (49.1%)		
Diploma or above	139 (51.9%)	185 (54.4%)			81 (46.8%)	88 (50.9%)		
Marital status			0.05	0.181			0.01	0.911
Single	112 (41.8%)	124 (36.5%)			62 (35.8%)	63 (36.4%)		
Married	156 (58.2%)	216 (63.5%)			111 (64.2%)	110 (63.6%)		
Living with older adults or children			0.01	0.818			0.03	0.568
No	97 (36.2%)	120 (35.3%)			55 (31.8%)	60 (34.7%)		
Yes	171 (63.8%)	220 (64.7%)			118 (68.2%)	113 (65.3%)		
Monthly income (in local currency MOP)			0.12	0.081			0.09	0.619
≤10,000	41 (15.3%)	57 (16.8%)			33 (19.1%)	30 (17.3%)		
10,001~20,000	97 (36.2%)	140 (41.2%)			61 (35.3%)	64 (37.0%)		
20,001~30,000	103 (38.4%)	95 (27.9%)			59 (34.1%)	55 (31.8%)		
30,001~40,000	17 (6.3%)	32 (9.4%)			11 (6.4%)	18 (10.4%)		
≧40,001	10 (3.7%)	16 (4.7%)			9 (5.2%)	6 (3.5%)		
Working industries			0.14	0.011			0.10	0.311
Travel agency	43 (16.0%)	82 (24.1%)			37 (21.4%)	39 (22.5%)		
Gaming	84 (31.3%)	83 (24.4%)			59 (34.1%)	55 (31.8%)		
Food and beverage	71 (26.5%)	107 (31.5%)			52 (30.1%)	42 (24.3%)		
Others	70 (26.1%)	68 (20.0%)			25 (14.5%)	37 (21.4%)		
COVID-19 vaccine uptake			0.31	<0.001			0.15	0.005
No	146 (54.5%)	82 (24.1%)			87 (50.3%)	61 (35.3%)		
Yes	122 (45.5%)	258 (75.9%)			86 (49.7%)	112 (64.7%)		

VA, vaccine attitude; SN, subjective norm; PBC, perceived behavioral control; CR, collective responsibility; PSM, propensity score matching; SMD, standardized mean difference.

**Table 3 vaccines-10-01295-t003:** The logistic regression analysis for COVID-19 vaccine uptake (*n* = 346).

Variables	Coefficient	S.E.	Wald	*p*-Value	OR	95% CI for OR	
						Lower	Upper
COVID-19 VA	0.139	0.029	23.304	<0.001	1.149	1.086	1.216
SN	0.012	0.007	2.868	0.090	1.012	0.998	1.027
PBC			32.892	<0.001			
PBC (moderate = 1)	−0.073	0.755	0.009	0.923	0.929	0.212	4.078
PBC (easy or very easy = 1)	1.729	0.737	5.510	0.019	5.636	1.330	23.882
CR (higher group = 1)	0.728	0.289	6.330	0.012	2.070	1.174	3.650
Zscore_COVID-19 VA × CR	−0.081	0.035	5.478	0.019	0.922	0.861	0.987
Working industries			3.212	0.360			
Working industries (travel agency = 1)	−0.750	0.566	1.757	0.185	0.472	0.156	1.432
Working industries (gaming = 1)	−0.254	0.434	0.342	0.558	0.776	0.331	1.816
Working industries (food and beverage = 1)	0.221	0.453	0.239	0.625	1.248	0.513	3.034
Age	0.004	0.018	0.039	0.844	1.004	0.968	1.041
Education (diploma or above = 1)	0.430	0.341	1.590	0.207	1.537	0.788	2.999
Monthly income	−0.041	0.154	0.073	0.787	0.959	0.710	1.296
Gender (female = 1)	−0.124	0.319	0.151	0.698	0.883	0.473	1.650
Marital status (married = 1)	0.475	0.334	2.027	0.155	1.609	0.836	3.095
Living with older adults or children (yes = 1)	−0.008	0.322	0.001	0.981	0.992	0.528	1.865
Constant	−2.459	1.325	3.443	0.064	0.085		

VA, vaccine attitude; SN, subjective norm; PBC, perceived behavioral control; CR, collective responsibility.

## Data Availability

All data generated or analyzed during this study are included in the published article and are available from the corresponding author upon reasonable request.

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
