# Peer review of "Higher Collective Responsibility, Higher COVID-19 Vaccine Uptake, and Interaction with Vaccine Attitude: Results from Propensity Score Matching"

_vaccines, 2022, doi:10.3390/vaccines10081295_

Round 1

Reviewer 1 Report

Dear Authors

It was with great pleasure that I reviewed your manuscript.

Here are some suggestions: 

I would like to check the KMO and total explained variance values regarding the Exploratory Factor Analysis.

I would also like to see if the items and scales follow a normal distribution or not.

Next to the interaction chart, there should also be a description of the results that it indicates.

My Best Regards

Reviewer 2 Report

The main purpose of this study was to determine the role of Collective Responsibility (C,R) in the decision of individuals to be vaccinated with the COVID19 vaccine.  The study was a survey of a limited population of workers in the tourist industry in the Macao Special Administration Region of China.  This is an interesting population with a high vaccination rate of 89.1%, so this was a good population to pose the survey questions as to the attitude of individuals to vaccination.

The authors had a sufficient number of respondents to the survey to allow for statistical analysis of the results.  Overall, the results are not surprising and predictable.  Not sure there were any results that the authors did not expect.  For example, individuals with a higher CR score are generally vaccinated.  Also those with a higher or lower CR score with no negative concerns with the vaccine are more likely to have positive vaccine uptake.  Some of the results verify or strengthen prior reports in the literature on the same subject of this study.

Overall, this is a reasonable attempt to understand the attitudes of a limited population of individuals related to COVID19 vaccines.  The authors note that this is the first study to examine the association of CR and COVID 19 vaccine uptake based on PSM.  The PSM method apparently allowed the authors to reduce the influence of other extraneous factors that might influence vaccine uptake, but are not statistically as relevant  as CR.

Finally, the authors suggest possible approaches that policy maker, health educators and practitioners might consider in communications relative to the value of vaccination for COVID19, and perhaps vaccination in general.

One comment, is the term vaccine uptake the appropriate wording?  When virologists think of vaccine uptake, it is usually in the since of uptake by immune cells or cells with receptors to the virus.  Perhaps this is the term used in other studies?  Does "vaccine uptake" = vaccinated?

Author Response

Response to Reviewer 2 Comments

Dear Reviewer,

Thank you very much for your insightful comments and suggestions on our manuscript entitled “Higher Collective Responsibility, Higher COVID-19 Vaccine Uptake, and Interaction with Vaccine Attitude: Results from Propensity Score Matching”.

Your feedback and suggestions are sincerely appreciated and highly valued. The following are point-by-point responses to your comments and concerns.

We are hopeful that our detailed responses have addressed your concerns. Meanwhile, we remain open to any further comments on the revised manuscript.

Sincerely,

Authors of vaccines-1833308

Point 1: The main purpose of this study was to determine the role of Collective Responsibility (C,R) in the decision of individuals to be vaccinated with the COVID19 vaccine.  The study was a survey of a limited population of workers in the tourist industry in the Macao Special Administration Region of China.  This is an interesting population with a high vaccination rate of 89.1%, so this was a good population to pose the survey questions as to the attitude of individuals to vaccination.

Response 1: Thank you for recognizing the purpose of this study and the sampling.

     Collective responsibility is an important perspective to explain motivations behind individuals’ vaccination behavior, especially for people holding a negative vaccine attitude. And the results might reflect one of the key reasons why Macao achieved such a high rate of vaccination.

     The participants were from six tourism satellite industries, which was a good sample because

  • it represented Macao’s economic and social characteristics as a tourist city,
  • collective responsibility was a very suitable factor in this population as these workers interacted with people more than others – so they were most impacted and influential during the pandemic, given about 7 million tourist arrivals in 2020 and more than 39 million in 2019.

Point 2: The authors had a sufficient number of respondents to the survey to allow for statistical analysis of the results.  Overall, the results are not surprising and predictable.  Not sure there were any results that the authors did not expect.  For example, individuals with a higher CR score are generally vaccinated.  Also those with a higher or lower CR score with no negative concerns with the vaccine are more likely to have positive vaccine uptake.  Some of the results verify or strengthen prior reports in the literature on the same subject of this study.

Response 2: Yes, most results are as we expected. Although some of previous studies have demonstrated the positive effect of collective responsibility on vaccination, this paper provided new insights rather than repeating findings in the literature.

1)                This paper dealt with potential confounding bias derived from individual differences in vaccine attitude, subjective norm and perceived behavioral control (see the discussion in Section 4.1, page 9, and results in Table 2, page 6-7). Using PSM, we reduced the confounding bias and ruled out alternative explanations. Hence, this study provided contribution with solid evidence and clearer findings, which reduced ambiguity in the literature.

2)                The second contribution and a converse part of our findings was a stronger effect of collective responsibility for people with negative vaccine attitude. Negative vaccine attitude, demonstrated by previous research and meta-analysis summaries, was seen an obstacle to vaccination – which was usually a challenge and a bad thing that made policy makers bite their nails; and some authorities responded with mandatory vaccination in spite of individual attitudes and encountered protests from people. Yet we found that collective responsibility could be a strategy to motivate these people and change their behavior. Empirically, the moderating results provided a new finding beyond the main effect of negative attitude in the past. And it would be the most unexpected part for practices, precisely targeting these people holding negative attitudes and a suitable use of collective responsibility for communication could make an effective strategy to promote vaccination – which would be a better way in line with values such as freedom and privacy.

In addition, we now have made clarification in the Limitation section of the manucript (Page 11, Line 388-396) with the extract as below:

“Thirdly, the case of Macao may not have high representativeness worldwide, since Macao is an Asian city with the highest population density in the world and its econ-omy heavily depends on tourism. Meanwhile, a USA-based study found that prosocial concerns promote vaccination against COVID-19 more in sparsely rather than densely populated areas [52]. The findings of this study should to be generalized with caution.”

Reference

  1. Jung, H. and D. Albarracín, Concerns for others increase the likelihood of vaccination against influenza and COVID-19 more in sparsely rather than densely populated areas. Proceedings of the National Academy of Sciences, 2021. 118(1).

Point 3: Overall, this is a reasonable attempt to understand the attitudes of a limited population of individuals related to COVID19 vaccines.  The authors note that this is the first study to examine the association of CR and COVID 19 vaccine uptake based on PSM.  The PSM method apparently allowed the authors to reduce the influence of other extraneous factors that might influence vaccine uptake, but are not statistically as relevant as CR.

Response 3: Yes, we completely agree with your opinion that “The PSM method apparently allowed the authors to reduce the influence of other extraneous factors that might influence vaccine uptake, but are not statistically as relevant as CR”.

Echoing your opinion, we have added a new paragraph in the Limitation section of the manuscript.

On Page 11, from Line 378-383:

“4.4. Limitations and Future Research

The present study had some limitations. Firstly, while generally consistent with randomized clinical trials when applied appropriately, the PSM method has its own limitations. Some data information may be lost during the process of propensity score matching because observations without suitable matches are excluded in the final estimation [50]. PSM method may help to alleviate the self-selection bias and the omitted variable bias, but can not fully address these concerns.”

Point 4: Finally, the authors suggest possible approaches that policy maker, health educators and practitioners might consider in communications relative to the value of vaccination for COVID19, and perhaps vaccination in general.

Response 4: Many thanks for your kind words. Indeed, we believe that communicating collective responsibility and motivating people to get vaccinated, rather than forcing people to do that, is a good approach. In this way, we can better protect individual freedom, privacy, other human rights and universal ethical principles.

Point 5: One comment, is the term vaccine uptake the appropriate wording?  When virologists think of vaccine uptake, it is usually in the since of uptake by immune cells or cells with receptors to the virus.  Perhaps this is the term used in other studies?  Does "vaccine uptake" = vaccinated?

Response 5:

Thank you very much for being rigorous on the terminology and we understand that scholars from different disciplines may have their own traditions on terms and definitions – we note that virology research discusses cellular uptake mechanisms/viral uptake into immune cells just as you mentioned.

When it is about human behavior of vaccination, however, “vaccine uptake” is defined as taking a dose of vaccine. It has the nuance of active intention and planned action, rather than a passive state imposed by “vaccinated”.

Currently, “vaccine uptake” (or vaccination uptake) and being/getting “vaccinated” have the same meaning and have been widely interchangeably used in the literature, when referring to vaccination behavior (evidence and details are listed below for your reference). Meanwhile, many scholars prefer to “vaccine uptake” due to its active nuance. For example, the WHO also uses the term of COVID-19 “vaccine uptake” (including “uptake of first dose and complete vaccination series”) and thereby calculates “vaccine uptake rates”.  

      Moreover, we were intended to use this term “vaccine uptake” for additional reasons:

  • It is a key variable and we name it “vaccine uptake” as a noun in our paper, while “vaccinated” (as an adjective) is usually used in terms such as vaccinated individuals/the vaccinated and be/get vaccinated in the literature. So “vaccine uptake” is more suitable to capture the concept and variable.
  • We investigated actual vaccination behavior, not just vaccination intention, and the term uptake could better reflect our contribution – given that Macao had a very high-level vaccination rate and it was an interesting and good population to study the proposed research questions (we thank your recognition in a comment above). Previous literature varied in variable measures when using the same word “vaccination” – some studies referred it to intention, while some others referred it to actual behavior. In this study, we differentiated willingness, planning, 1 dose and 2 doses using an established way of measurement.

Evidence and details:

  • With the same meaning, 3 references cited in this paper entitled “vaccine uptake” (Kowk et al., 2021; Nicholls et al., 2021; Fan et al., 2021), while another 3 used “vaccinated” in title (Reiter, Pennell & Katz, 2020; Dorman et al., 2021; Willis et al., 2022). So either term was a common expression and they were used interchangeably in these papers.
  • In more papers, these two terms did mean the same and those researchers used both terms interchangeably in their writings.
  • Searching titles on Google Scholar (5 recent years): vaccine uptake (1310 results), vaccination uptake (846 results), a total of 2156 results excluding other terms containing uptake; vaccinated (3720 results).
  • Searching papers published in Vaccines: vaccine uptake (112 results), vaccination uptake (75 results), a total of 187 results excluding other terms containing uptake; vaccinated (210 results).
  • World Health Organization. Monitoring COVID-19 vaccine uptake. Available online: https://www.who.int/europe/activities/monitoring-covid-19-vaccine-uptake (accessed on 2 August 2022).